# A Simple and Effective Phosphine-Doping Technique for Solution-Processed Nanocrystal Solar Cells

**DOI:** 10.3390/nano13111766

**Published:** 2023-05-30

**Authors:** Chenbo Min, Yihui Chen, Yonglin Yang, Hongzhao Wu, Bailin Guo, Sirui Wu, Qichuan Huang, Donghuan Qin, Lintao Hou

**Affiliations:** 1School of Materials Science and Engineering, South China University of Technology, Guangzhou 510640, China; m13395297158@163.com (C.M.); 13828224792@163.com (Y.C.); wbxdslyyl@163.com (Y.Y.); wuhzsci@163.com (H.W.); 13617228755@163.com (B.G.); nssquinque@163.com (S.W.); hqc666666nb@163.com (Q.H.); 2State Key Laboratory of Luminescent Materials & Devices, Institute of Polymer Optoelectronic Materials & Devices, South China University of Technology, Guangzhou 510640, China; 3Guangdong Provincial Key Laboratory of Optical Fiber Sensing and Communications, Guangzhou Key Laboratory of Vacuum Coating Technologies and New Energy Materials, Siyuan Laboratory, Department of Physics, Jinan University, Guangzhou 510632, China; thlt@jnu.edu.cn

**Keywords:** cadmium telluride nanocrystals, photovoltaic device, phosphine-doping technique

## Abstract

Solution-processed cadmium telluride (CdTe) nanocrystal (NC) solar cells offer the advantages of low cost, low consumption of materials and large-scale production via a roll-to-roll manufacture process. Undecorated CdTe NC solar cells, however, tend to show inferior performance due to the abundant crystal boundaries within the active CdTe NC layer. The introduction of hole transport layer (HTL) is effective for promoting the performance of CdTe NC solar cells. Although high-performance CdTe NC solar cells have been realized by adopting organic HTLs, the contact resistance between active layer and the electrode is still a large problem due to the parasitic resistance of HTLs. Here, we developed a simple phosphine-doping technique via a solution process under ambient conditions using triphenylphosphine (TPP) as a phosphine source. This doping technique effectively promoted the power conversion efficiency (PCE) of devices to 5.41% and enabled the device to have extraordinary stability, showing a superior performance compared with the control device. Characterizations suggested that the introduction of the phosphine dopant led to higher carrier concentration, hole mobility and a longer lifetime of the carriers. Our work presents a new and simple phosphine-doping strategy for further improving the performance of CdTe NC solar cells.

## 1. Introduction

A major obstacle for cadmium telluride (CdTe) in facilitating high-performance photovoltaic devices is the high contact resistance associated with the metal due to the high electron affinity of CdTe [1,2,3,4,5]. Solution-processed CdTe NC solar cells inherit this problem from their evaporated counterparts and require refinement. Prior studies have suggested that the introduction of a hole transport layer (HTL) can relieve this problem, especially for CdTe solar cells adopting an inverted structure. In this case, an HTL is inserted between the active CdTe layer and the back-contact electrodes [6,7,8,9,10]. In such circumstances, the HTL can mediate the charge extraction process and enhance the performance of solar cells. In order to minimize recombination of the carrier at the semiconductor–metal interface, choosing HTLs with a high work function is crucial. By adopting Spiro (2,2′,7,7′-tetrakis[*N*,*N*-di(4-methoxyphenyl)amino]-9,9′-spirobifluorene) as an HTL and using CdTe NC solar cells with a configuration of ITO (indium tin oxide)/TiO_2_/CdTe/Spiro/Au, the devices showed enhanced performance, as a dipole layer formed on the CdTe NC–HTL interface and strengthened the built-in electric field, leading to the enhanced performance of the solar cells [11,12]. Later on, other HTLs such as cross-linkable conjugated polymer poly(diphenylsilane-co-4-vinyl-triphenylamine) and poly(phenylphosphine-co-4-vinyl-triphenylamine) [13,14] were applied in CdTe NC solar cells, and the power conversion efficiency (PCE) of CdTe solar cells was increased by up to 9%. Despite its promise, organic HTLs still face a challenge with contact resistance due to the high resistance of organic materials. This results in a portion of the current being blocked by the parasitic resistance of the organic HTL.

Vacuum thermal evaporated CdTe thin-film solar cells use inorganic HTLs to solve this problem. By evaporating a thin layer of Cu (~1 nm) directly after the deposition of CdTe thin-film, a Cu-doped CdTe layer can be formed. This layer can serve as an HTL and increase the concentration of holes and prolong the carrier’s lifetime [15,16]. Although the copper-doping technique is commercially successful, it is still not ideal, since excessive Cu can diffuse into the active CdTe layer along the grain boundaries [17,18]. For solution-processed CdTe NC solar cells, the abundant crystal boundaries within the active CdTe NC layer are destructive for the copper-doping technique which has been successfully used in cadmium telluride thin film cells prepared by vacuum technics, thus requiring polishing/refining of the active layer to minimize the negative impact on the cells’ performance. Cu may easily diffuse into the active CdTe NC layer, leading to inversion of the *p*–*n* junction [19,20]. Recent studies have suggested that doping single crystalline CdTe with Group V elements can prevent this problem and improve the open-circuit voltage (*V*_oc_) of the CdTe solar cells by up to 1 V [21,22]. In addition, the phosphine-doping technique can suppress the compensation effect and the recombination of the interface as well. To date, there are still few reports on doping CdTe NC solar cells prepared by a solution process with Group V elements, since it is still difficult to fabricate homogeneous active doped layers of the element in solution-processed CdTe NC solar cells. In this work, by doping the active CdTe NC layer with phosphine salt, we managed to eliminate the uncontrollable diffusion of the dopant into the active layer and successfully fabricated a phosphine-doped layer [23,24,25]. Using triphenylphosphine (TPP) as a phosphine source, we doped our active CdTe NC layer via a simple solution process. By controlling the concentration of TPP in anhydrous methanol, the diffusion of phosphine in the active CdTe NC layer was controlled. The statistical analysis of the CdTe NC thin film with or without TPP dopant showed that greater hole mobility can be obtained by doping NC with TPP. Moreover, NC solar cells with TPP showed better *p–n* junction quality, a lower back-contact energy barrier, a longer carrier lifetime and a of low concentration of defects compared with devices without TPP doping. After optimization of the fabrication process, phosphine-doped solar cells exhibited a high PCE of 5.41%, which is ~30% higher than the dopant-free device (4.05%).

## 2. Experimental Procedure

The zinc oxide (ZnO) precursor, cadmium selenide (CdSe) and CdTe NCs were synthesized by following previously reported methods [14]. TPP was selected as the phosphine source. According to the solution process, TPP was dissolved into methanol to form a TPP–methanol solution with different concentrations. CdTe NCs were dispersed into pyridine and a 1-PrOH compound (with a volume ratio of 1:1) at a concentration of 45 mg/mL, while CdSe NCs were dispersed into pure pyridine at a concentration of 30 mg/mL. All the NC dispersions were filtered through a 0.45 μm filter before further usage. The ZnO precursor was prepared by dissolving zinc acetate hydrate and ethanolamine into 2-methoxymethanol solution at a concentration of 0.5 mol/L. The NC thin films were deposited via a layer-by-layer annealing process as previously reported [14], and a multilayered ITO/ZnO/CdSe NC/CdTe NC was fabricated by this means. The CdCl_2_ treatment was necessary to make the grain grow after each layer of CdTe spin coating, except for the last layer. Upon the fabricated structure, TPP–methanol solutions with different concentrations (1000 ppm, 2000 ppm, 3000 ppm and 5000 ppm) were deposited on the substrate and spin-casted at 2500 rpm for 15 s, while a TPP-free sample was prepared as the control sample. The samples were put on top of a hotplate and annealed at different temperatures from 330 °C to 350 °C for 30 min. Finally, 80 nm of Au was deposited on the ITO/ZnO/CdSe NC/CdTe NC with or without a TPP substrate as the back-contact electrode via vacuum thermal evaporation. The active area of the as-fabricated solar cells was 0.16 cm^2^.

## 3. Results and Discussion

Characterization via atomic force microscopy (AFM) was carried out to profile the morphology of CdTe NC thin films with or without TPP. The AFM image of the control sample is presented in Figure 1a and those of the CdTe NC thin film with TPP are presented in Figure 1b–d. The TPP-free CdTe NC thin film exhibited a relatively rougher surface morphology with a root mean square (RMS) of roughness of about 10.8 nm (Figure 1a). The particle size (~150 nm) at the surface was larger compared with the NCs in dispersion form due to the growing process of CdCl_2_ grains. The introduction of TPP may have been beneficial for the formation of a flattened surface. The CdTe NC thin-films with TPP appeared smoother compared with the control sample. The RMS of roughness of the CdTe NC thin film coupled with TPP–methanol solutions of 1000 ppm, 2000 ppm and 5000 ppm were 3.57 nm, 4.02 nm and 3.23 nm, respectively (Figure 1b–d). A smoother surface may lead to a higher carrier collection rate due to the reduction in surface defects. It is obvious that after the introduction of TPP without CdCl_2_ during the annealing of the CdTe NC thin film, the size of the particles was significantly reduced, but the distribution was uniform. Chloride ions can promote the growth of cadmium tellurite grains, which has been confirmed in the literature [1]. We assumed that the TPP would accumulate at the grain boundary and form a Cd–P bond, reducing the inter-grain space and leading to increased flatness. To examine our assumption, we used X-ray photoelectron spectroscopy (XPS) characterization to probe the Cd–P bond, and the result is presented in Figure 1e,f and Appendix A. The elemental characteristic peaks of Cd, Te, O, C and P for the CdTe NC thin film with or without TPP could be observed in the XPS spectra. It was noted that the P content in the CdTe NC thin film processed by doping with 5000 ppm TPP was almost nine times higher than that of the TPP-free process, confirming the effective doping of phosphine using the TPP–methanol solution. As phosphine salt (TPP) was applied as the phosphine source, the diffusion of phosphine in the active layer was restricted.

To investigate the variation in the optical properties of CdTe NC thin film with TPP, three layers of CdTe NC were deposited on the ITO substrate and treated with under same conditions. Figure 2a,b exhibit the transmission spectra and the Tauc plot [26] of the CdTe NC thin films. An ITO substrate was selected as the standard sample in the characterization. In Figure 2a, the ITO/CdTe thin films with or without TPP block the light in the whole visible spectrum and the values of transmission increase with the wavelength due to inadequate absorption (the thickness of the CdTe NC was ~240 nm). As shown in Figure 2b, the bandgap of the CdTe NC with and without P doping was ~1.50 eV and ~1.47 eV (the intercept of the blue line on the x-axis), respectively. The band gap was obtained by taking the intercept of the fitted line. We can see that phosphine doping had little effect on the band gap of the CdTe NC thin film, consistent with prior studies [21].

Phosphine-doped CdTe thin films proved to be a good strategy for eliminating the energy loss at the back contact and improved the *V*_oc_ of CdTe thin-film solar cells. Figure 3a shows the schematic diagram of our CdTe solar cell with an inverted structure. Typically, the devices consist of a 40 nm ZnO window layer, a 60 nm CdSe NC electron transport layer and a ~500 nm active CdTe NC layer. Figure 3b shows the band alignment diagram of the as-fabricated NC solar cells [11,13,21]. As phosphine is a dopant with a shallow energy level, its energy level is close to the valence band, which can boost the voltage output of the devices. To investigate the effects of TPP content on the devices’ performance, solar cells processed with different concentrations of a TPP–methanol solution were fabricated. The current density–voltage (*J*–*V*) curves of NC solar cells processed with different TPP–methanol solutions are presented in Appendix A, while the performance of the corresponding devices are summarized in Appendix A and Figure 3c. The increases in PCE were from 2.31% to 4.38% when the TPP concentration increased from 1000 ppm to 2000 ppm. Further increases in the TPP concentration led to degradation of the PCE and only 1.49% of the PCE was obtained for the device processed with the 5000 ppm TPP–methanol solution. The device’s performance reduces with TPP concentrations higher or lower than 2000 ppm, which was primarily attributed to the reduced short-circuit current (*J*_sc_) and *V*_oc_. We speculated that at low TPP concentrations, the inadequate coverage of the NC’s surface may produce insufficient passivation of the interface, thus failing to form an effective phosphine-doped layer. On the contrary, for the TPP–methanol solution with a high TPP concentration, redundant TPP may form a defective center and trap photogenerated carriers, leading to degradation of the device’s performance. Therefore, in all of the devices discussed below, we chose to adopt 2000 ppm TPP as the optimal phosphine-doping condition. The annealing treatment also has important effects on the grains’ growth and the dopant diffusion conditions. The optimal annealing condition for the device was annealing at 340 °C for 30 min (as shown in Figure 3d, Appendix A). Figure 3e presents the *J*–*V* curves of the best devices with and without phosphine doping, and the corresponding performance is summarized in Table 1. The best device with phosphine doping exhibited an enhanced *V*_oc_ of 0.56 V, a *J*_sc_ of 18.19 mA/cm^2^ and a fill factor (FF) of 53.16%, and the control device exhibited a *V*_oc_ of 0.52 V, a *J*_sc_ of 17.52 mA/cm^2^ and an FF of 44.48%, leading to a PCE of 5.41% and 4.05%, respectively. This enhancement may be mainly attributed to the low defects and the low carrier recombination rate at the semiconductor–metal interface of the phosphine-doped device. The corresponding external quantum efficiency (EQE) spectra for the devices with and without TPP doping are shown in Figure 3f. The phosphine-doped device exhibited higher EQE values across almost all the visible wavelength range, compared with the control sample. By integrating the EQE data, a *J*_sc_ of 17.39 and 16.15 mA cm^−2^ were obtained, close to the data measured in the *J*–*V* characterization. The enhanced EQE values may be attributed to the reduced carrier recombination and increased mobility of holes at the semiconductor–metal interface.

To investigate the effect of our phosphine-doping technique on the charge transport properties of CdTe NC thin films, we used the space charge–limited current (SCLC) method to qualitatively measure the carrier mobility of the holes of CdTe NC thin films. The carrier mobility can be calculated via the following equation [27]
$$J=\frac{9}{8}\frac{\varepsilon_{0}\varepsilon_{r}\mu_{p}{\left(V-V_{bi}-V_{s}\right)}^{2}}{L^{3}}$$
where $\varepsilon_{0}$ is the permittivity of free space, $\varepsilon_{r}$ is the relative dielectric constant of CdTe, *L* is the thickness of the CdTe NC film, *μ*_p_ is the holes’ mobility, *V* is the applied voltage and *V*_s_ is the voltage drop due to contact resistance and *V*_bi_ is the built-in voltage. As shown in Figure 4a–e and Table 2, the mobility of CdTe NC thin films with TPP doping at concentrations of x = 0, 1000, 2000, 3000 and 5000 ppm was 2.1 × 10^−4^, 3.46 × 10^−4^, 3.65 × 10^−4^, 3.18 × 10^−4^ and 2.89 × 10^−4^ cm^2^/Vs, respectively. It is obvious that all phosphine-doped CdTe NC thin films showed greater hole mobility than the control device. The *μ*_p_ increased almost linearly with the TPP doping concentration from 0 to 2000 ppm and dropped when the doping concentration increased further. The highest *μ*_p_ 2qw obtained in the case of samples doped at 2000 ppm. Greater hole mobility reduced the recombination of the interface at the back contact and resulted in better device performance, which conformed with the results of the *J*–*V* curves.

The built-in electric field formed by alignment of the energy band determines the *V*_oc_ of a photovoltaic device. Through capacitance–voltage (*C*–*V*) characterization in a dark environment, we can investigate the change in the built-in electric field (*V*_bi_) of the devices with and without phosphine doping. The *C*–*V* measurement was carried out with an increase in the bias voltage at a constant frequency of 1000 Hz. The *V*_bi_ can be calculated by the Mott–Schottky equation [28]
$$C^{-2}=\frac{2}{A^{2}q\varepsilon_{0}\varepsilon N_{A}}\left(V_{bi}-V\right)$$
where *C* is the depletion of the layer’s capacitance, *A* is active area, *ε* is the relative dielectric constant (10.6 for CdTe), *ε*_0_ is the permittivity in a vacuum and *N*_A_ is the net concentration of acceptors. The *V*_bi_ of NC devices was extracted at a forward bias from the slope. As shown in Figure 5a, the *V*_bi_ of the NC solar cells with 2000 ppm, 5000 ppm and without TPP doping was 0.66 V, 0.59 V and 0.52 V, respectively. It is worth noting that the absolute slope of the fitted line (the purple line from Figure 5a) was the least at 2000 ppm, which means that the concentration of holes was the highest, according to the Mott–Schottky equation. The improvement in *V*_bi_ for phosphine-doped devices may be attributed to the low contact resistance of the CdTe NC/Au interface; in other words, greater hole transportation efficiency was realized by phosphine doping. In addition, the P–Cd bond may passivate the interface’s defects and form a dipole layer, leading to an increase in *V*_bi_. The *J*–*V* curves in the dark for devices with and without TPP are presented in Figure 5b. We can observe that the reverse dark current of the phosphine-doped NC solar cell was significantly lower than that of the control device. The lower dark current implies the reduction in the carrier’s recombination at the back contact, resulting in improvements in *J*_sc_ and the device’s performance. The effect of TPP on the lifetime charge recombination of the NC solar cells was investigated by transient photovoltage (TPV). As shown in Appendix A, the control device without TPP doping shows the shortest charge recombination lifetime of 0.82 μs. On the other hand, devices with 2000 ppm and 5000 ppm TPP doping showed longer charge recombination lifetimes of 1.51 μs and 1.24 μs, which suggested that the charge recombination was low for the TPP-doped device.

The stability of NC solar cells is an important issue for real-world applications. All the devices were unencapsulated and stored in ambient conditions for the stability test. As shown in Figure 5c,d, all the devices were extremely stable during storage and measurement. The control device maintained an average of 97.5% of the original PCE in all stages, while the TPP-doped devices maintained an average 99.2% of the original PCE. As the whole active layer consists of inorganic semiconductor materials (ZnO, CdSe NC and CdTe NC), these materials have been confirmed to be stable and are ready for real-world application. In the case of phosphine-doped CdTe NC solar cells, the diffusion of phosphine was restricted, as we adopted phosphine salt instead of elemental phosphine as the phosphine source.

In conclusion, we have demonstrated the use of a simple phosphine salt (TPP) as phosphine source for phosphine-doped CdTe NC solar cells. The established phosphine-doped active layer increased the built-in potential and improved the carrier’s collection ratio at the back contact interface, resulting in low recombination losses. By optimizing the doping conditions, simultaneous enhancements in *V*_oc_, *J*_sc_ and FF were realized, which resulted in the improvement in PCE. On the other hand, the phosphine-doped NC solar cells exhibited excellent stability, maintaining 99.2% of the original PCE after exposure to ambient conditions for 60 days. We believe that the performance of TPP-doped devices could be further improved in the future via optimizing the technology of preparing the devices, the devices’ architecture et al. Our work has developed an effective way to further improve the CdTe NC device’s performance by incorporating a phosphine-salt-doped active layer.

## Figures and Tables

**Figure 1 nanomaterials-13-01766-f001:**
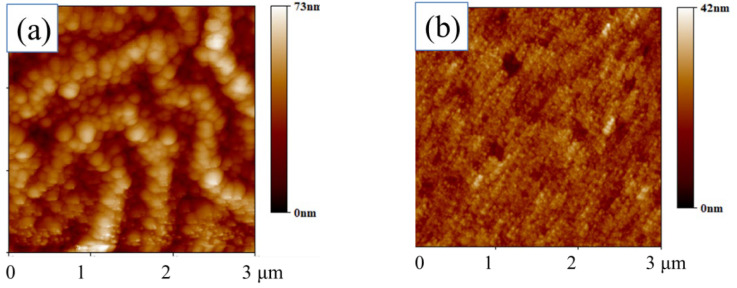
AFM images of ITO/ZnO/CdTe with TPP doped at different concentrations: (**a**) 0 ppm, (**b**) 1000 ppm, (**c**) 2000 ppm, (**d**) 5000 ppm. (**e**) XPS survey scan of CdTe NC thin films with or without TPP doping; (**f**) narrow XPS scan for P_2p_.

**Figure 2 nanomaterials-13-01766-f002:**
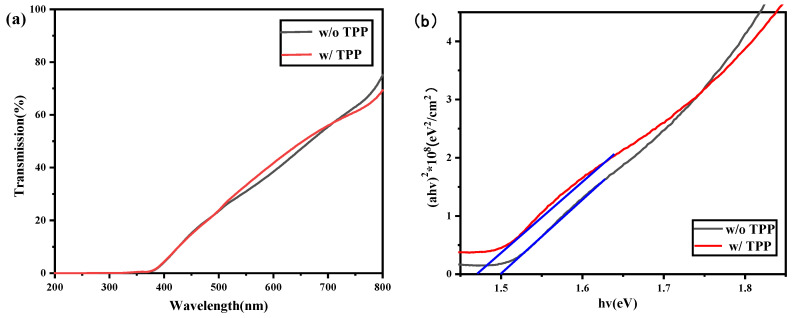
(**a**) Transmission spectra of CdTe NC thin films with or without TPP doping. (**b**) Tauc plots of the (*αhv*)^2^ versus the photon energy (*hv*) of the CdTe NC thin films with or without TPP doping.

**Figure 3 nanomaterials-13-01766-f003:**
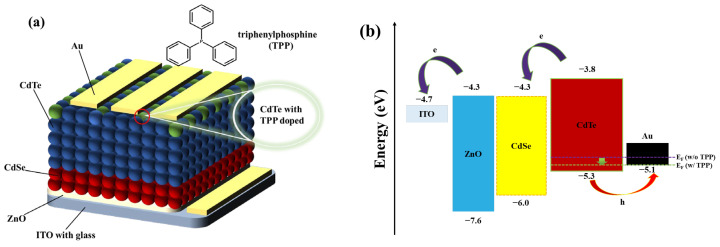
(**a**) A schematic of the CdTe/CdSe solar cells with an inverted structure and the chemical structure of TPP. (**b**) The energy levels of ITO, ZnO, CdSe, CdTe (without TPP) and Au. (**c**) Variation in the PCE of CdSe/CdTe NC solar cells with different TPP concentrations (device structure: ITO/ZnO/CdSe/CdTe/TPP/Au). (**d**) Variation in the PCE of CdSe/CdTe NC solar cells with different annealing temperatures (the *J*–*V* curves and parameters are presented in Appendix A). (**e**) *J*–*V* characteristics of the best performing devices (device structure: ITO/ZnO/CdSe/CdTe/TPP (without Au) under AM1.5 G illumination and (**f**) the corresponding EQE spectra.

**Figure 4 nanomaterials-13-01766-f004:**
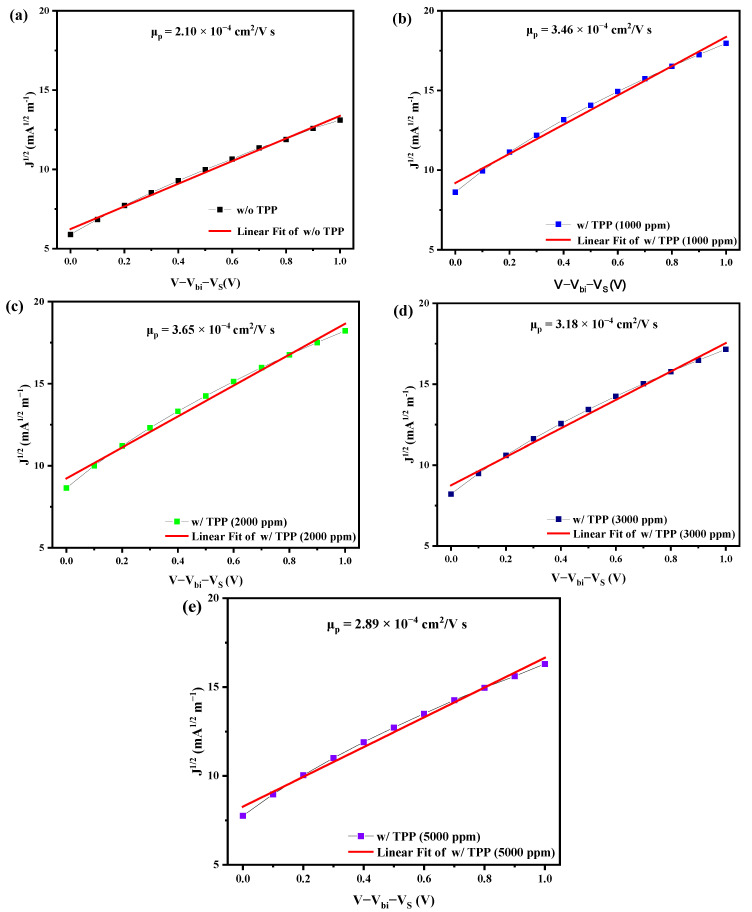
Linear fitting of the SCLC measurements of CdTe NC thin films with different TPP doping concentrations, namely (**a**) without TPP, (**b**)1000 ppm, (**c**) 2000 ppm, (**d**) 3000 ppm and (**e**) 5000 ppm, where ε_0_ = 8.85 × 10^−12^, ε_r_ = 10, *L* = 160 nm and *V*_bi_ + *V*s = 0.3 V.

**Figure 5 nanomaterials-13-01766-f005:**
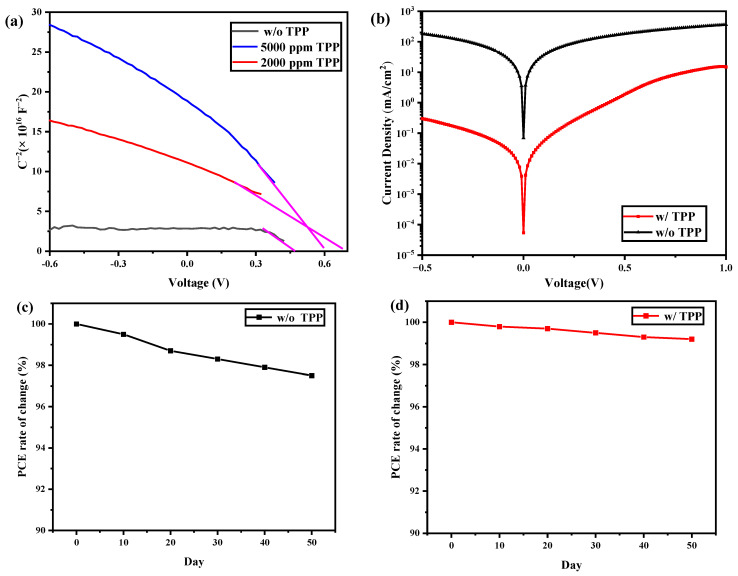
(**a**) *C*–*V* characteristics of devices with 2000 ppm, 5000 ppm and without TPP. (**b**) *J*–*V* characteristics of CdSe/CdTe NC solar cells with or without TPP (device structure: ITO/ZnO/CdSe/CdTe/TPP (without)/Au) under dark conditions. (**c**) Evolution of PCE for devices without TPP under ambient conditions. (**d**) Evolution of PCE for devices with TPP under ambient conditions.

**Table 1 nanomaterials-13-01766-t001:** Summary of the photovoltaic parameters of the best NC solar cells without TPP doping.

Architecture of the Device	*V*_oc_ (V)	*J*_sc_ (mA/cm^2^)	FF (%)	PCE (%)	*R*_s_ (Ω·cm^2^)	*R*_sh_ (Ω·cm^2^)
ITO/ZnO/CdSe/CdTe/Au	0.52	17.52	44.48	4.05	24.06	92.35
ITO/ZnO/CdSe/CdTe/TPP/Au	0.56	18.19	53.16	5.41	8.14	182.87

**Table 2 nanomaterials-13-01766-t002:** Hole mobility of CdTe thin films with different TPP doping concentrations.

Sample	Hole Mobility (cm^2^ v^−1^ s^−1^)
Without TPP	2.10 × 10^−4^
1000 ppm	3.46 × 10^−4^
2000 ppm	3.65 × 10^−4^
3000 ppm	3.18 × 10^−4^
5000 ppm	2.89 × 10^−4^

## Data Availability

No new data was created.

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
