# Peer review of "A Simple and Effective Phosphine-Doping Technique for Solution-Processed Nanocrystal Solar Cells"

_nanomaterials, 2023, doi:10.3390/nano13111766_

Round 1

Reviewer 1 Report

The manuscript titled "A simple and effective phosphine-doping technique for solution-processed nanocrystal solar cells", by Min et al. reports on the solution-processed cadmium telluride (CdTe) nanocrystal (NC) based solar cells that offer several advantages including cost-effectiveness, low consumption materials and large scale production via R2R manufacture process. While pristine and unmodified CdTe NC solar cells show inferior performance due to crystal boundaries within the CdTe NC active layer, the introduction of a hole transport layer (HTL) is an effective way to promote device performance. However, the contact resistance between the active layer and electrode due to the parasitic resistance of HTL is still a challenge that the authors attempted to address. They have conducted an annealing and inert atmosphere-free phosphine doping using triphenylphosphine as a phosphine source. The authors found a significant increase in power conversion efficiency of the device to 5.41% and in extraordinary stability. The authors attribute phosphine dopant leads to higher carrier (hole) concentration as well as mobility. Despite their findings, the current work appears to be more of data reporting and lacks scientific novelty. Additional experiments must be done to support their experimental data that may help definitive insights into the findings. Moreover, claiming the current work i.e., NC materials processed by solution for photovoltaic devices for industrial application is somewhat an exaggeration. Based on these considerations, the paper in its current form must be rejected.

The manuscript titled "A simple and effective phosphine-doping technique for solution-processed nanocrystal solar cells", by Min et al. reports on the solution-processed cadmium telluride (CdTe) nanocrystal (NC) based solar cells that offer several advantages including cost-effectiveness, low consumption materials and large scale production via R2R manufacture process. While pristine and unmodified CdTe NC solar cells show inferior performance due to crystal boundaries within the CdTe NC active layer, the introduction of a hole transport layer (HTL) is an effective way to promote device performance. However, the contact resistance between the active layer and electrode due to the parasitic resistance of HTL is still a challenge that the authors attempted to address. They have conducted an annealing and inert atmosphere-free phosphine doping using triphenylphosphine as a phosphine source. The authors found a significant increase in power conversion efficiency of the device to 5.41% and in extraordinary stability. The authors attribute phosphine dopant leads to higher carrier (hole) concentration as well as mobility. Despite their findings, the current work appears to be more of data reporting and lacks scientific novelty. Additional experiments must be done to support their experimental data that may help definitive insights into the findings. Moreover, claiming the current work i.e., NC materials processed by solution for photovoltaic devices for industrial application is somewhat an exaggeration. The English must be edited occasionally and there are numerous instances to list them here. Based on these considerations, the paper in its current form must be rejected.

Author Response

Response to the referee’s report

nanomaterials- 2364988

(Italic character- Referee’s comment; red character-author’s response to the referee)

Reviewer #1:

Comments and Suggestions for Authors

The manuscript titled "A simple and effective phosphine-doping technique for solution-processed nanocrystal solar cells", by Min et al. reports on the solution-processed cadmium telluride (CdTe) nanocrystal (NC) based solar cells that offer several advantages including cost-effectiveness, low consumption materials and large scale production via R2R manufacture process. While pristine and unmodified CdTe NC solar cells show inferior performance due to crystal boundaries within the CdTe NC active layer, the introduction of a hole transport layer (HTL) is an effective way to promote device performance. However, the contact resistance between the active layer and electrode due to the parasitic resistance of HTL is still a challenge that the authors attempted to address. They have conducted an annealing and inert atmosphere-free phosphine doping using triphenylphosphine as a phosphine source. The authors found a significant increase in power conversion efficiency of the device to 5.41% and in extraordinary stability. The authors attribute phosphine dopant leads to higher carrier (hole) concentration as well as mobility. Despite their findings, the current work appears to be more of data reporting and lacks scientific novelty. Additional experiments must be done to support their experimental data that may help definitive insights into the findings.

Thanks for the comments. The phosphine or other V group element doped CdTe thin film can decrease the energy loss at the interface of CdTe /contact metal and improve the open circuit voltage of thin film solar cells, which had been confirmed in the previous reports (Reference 21, 22). However, there is still few reports on the phosphine doped CdTe NC with phosphine salt via a simple solution process. In this article, based on our previous work, we first time presented TPP as phosphine salt for solution processed CdTe NC solar cells. We proved that the NC solar cells performance can be significantly improved. Based on the XPS, SCLC and C-V measurement, we then inference that the TPP doping can increase the carriers mobility and reinforce the build in field, which lead to high device performance. In order to further clarify this, The effect of TPP on the charge recombination lifetime of NC solar cells is investigated by transient photovoltage (TPV). As shown in Figure S3, controlled device without TPP doped shows the lowest charge recombination lifetime of 0.82 μs. On the other hand, devices with 2000 ppm and 5000 ppm TPP doped show higher charge recombination lifetime of 1.51 μs and 1.24 μs, which suggests that the charge recombination is low for the TPP doped device. We have presented the detail discussion in the revised manuscript.

  1. Moreover, claiming the current work i.e., NC materials processed by solution for photovoltaic devices for industrial application is somewhat an exaggeration. Based on these considerations, the paper in its current form must be rejected.

Thanks for the comments. We have revised the expression in the revised manuscript. We have asked a colleague to correct the grammatical errors throughout the text.

Reviewer 2 Report

After carefully checking the manuscript entitled: "A simple and effective phosphine-doping technique for solution-processed nanocrystal solar cells", I can mention that the contribution is interesting, because it reveals an interesting processing method of chalcogenide based active layers to increase the performance of solar cells. I recommend this manuscript for publication after addressing the following concerns:

1. How does the TPP afect the particle size distribution of the chalconide nanocrystals? explain.

2. What is the main chemical interaction between the TPP and the nanocrystals. Is it a surface interaction? Are there passivation phenomena in the photomaterial? explain.

3. How were the energy levels of ITO, ZnO, CdSe, CdTe (w/o TPP) and Au determined in the Figure 3b? please, explain.

some changes in english should be observed in the main text

Author Response

Response to the referee’s report

nanomaterials- 2364988

(Italic character- Referee’s comment; red character-author’s response to the referee)

Reviewer #2:

After carefully checking the manuscript entitled: "A simple and effective phosphine-doping technique for solution-processed nanocrystal solar cells", I can mention that the contribution is interesting, because it reveals an interesting processing method of chalcogenide based active layers to increase the performance of solar cells. I recommend this manuscript for publication after addressing the following concerns:

  1. How does the TPP afect the particle size distribution of the chalconide nanocrystals? explain.

Thanks for the comments. We have explained the effects of TPP on the NCs particle size distribution in the revised manuscript.

  1. What is the main chemical interaction between the TPP and the nanocrystals. Is it a surface interaction? Are there passivation phenomena in the photomaterial? explain.

Thanks for the comments. The interaction between the TPP and the nanocrystals is the formation of Cd-P bond between the CdTe NC and the TPP. Yes, this is a surface interaction as phosphine salt (TPP) not phosphine is applied as phosphine sources, the diffusion of phosphine in the active layer will be restricted. Therefore the passivation is not occurs in the active layer. We have explained in the revised manuscript.

  1. How were the energy levels of ITO, ZnO, CdSe, CdTe (w/o TPP) and Au determined in the Figure 3b? please, explain.

Thanks for the comments. The energy levels of ITO, ZnO, CdSe, CdTe can be found in the previous reports (reference 11, 13).